# The Effects of Device-Based Cardiac Contractility Modulation Therapy on Left Ventricle Global Longitudinal Strain and Myocardial Mechano-Energetic Efficiency in Patients with Heart Failure with Reduced Ejection Fraction

**DOI:** 10.3390/jcm11195866

**Published:** 2022-10-04

**Authors:** Daniele Masarone, Michelle M. Kittleson, Stefano De Vivo, Antonio D’Onofrio, Ernesto Ammendola, Gerardo Nigro, Carla Contaldi, Maria L. Martucci, Vittoria Errigo, Giuseppe Pacileo

**Affiliations:** 1Heart Failure Unit, Department of Cardiology, AORN dei Colli-Monaldi Hospital, 80131 Naples, Italy; 2Department of Cardiology, Smidt Heart Institute, Cedars-Sinai, Los Angeles, CA 90048, USA; 3Electrophysiology Unit, Department of Cardiology, AORN dei Colli Monaldi Hospital, 80131 Naples, Italy; 4Cardiology Unit, Department of Medical Translational Sciences, University of Campania “Luigi Vanvitelli”, 80131 Naples, Italy

**Keywords:** cardiac contractility modulation, heart failure with reduced ejection fraction, global longitudinal strain, myocardial mechano-energetics efficiency

## Abstract

Background: Virtually all patients with heart failure with reduced ejection fraction have a reduction of myocardial mechano-energetic efficiency (MEE). Cardiac contractility modulation (CCM) is a novel therapy for the treatment of patients with HFrEF, in whom it improves the quality of life and functional capacity, reduces hospitalizations, and induces biventricular reverse remodeling. However, the effects of CCM on MEE and global longitudinal strain (GLS) are still unknown; therefore, this study aims to evaluate whether CCM therapy can improve the MEE of patients with HFrEF. Methods: We enrolled 25 patients with HFrEF who received an Optimizer Smart implant (the device that develops CCM therapy) between January 2018 and January 2021. Clinical and echocardiographic evaluations were performed in all patients 24 h before and six months after CCM therapy. Results: At six months, follow-up patients who underwent CCM therapy showed an increase of left ventricular ejection fraction (30.8 ± 7.1 vs. 36.1 ± 6.9%; *p* = 0.032) as well a rise of GLS 10.3 ± 2.7 vs. −12.9 ± 4.2; *p* = 0.018), of MEE (32.2 ± 10.1 vs. 38.6 ± 7.6 mL/s; *p* = 0.013) and of MEE index (18.4 ± 6.3 vs. 24.3 ± 6.7 mL/s/g; *p* = 0.022). Conclusions: CCM therapy increased left ventricular performance, improving left ventricular ejection fraction, GLS, as well as MEE and MEEi.

## 1. Introduction

Myocardial mechano-energetic efficiency (MEE) expresses the heart’s ability to convert adenosine triphosphate (ATP), obtained from aerobic metabolism, into mechanical work [1]. Increased energy dissipation is a pathophysiologic hallmark of heart failure (HF) with reduced ejection fraction (HFrEF), in which MEE is reduced [2]. Although the gold standard for quantification of MEE is cardiac catheterization (bilateral and of the coronary sinus) [3], recently, an echocardiographic approach has been proposed, enabling more extensive clinical applications [4,5]. Cardiac contractility modulation (CCM) is an innovative therapy for the treatment of patients with HF [6] that through delivery, via an implantable device (Optimizer Smart^®^, Impulse Dynamics, Marlton, NJ, USA), of high-energy biphasic non-excitatory impulses during the absolute refractory period of the cardiomyocytes results in improved calcium handling [7], reverses titin downregulation and fetal gene expression [8,9] and reduces adrenergic tone and myocardial fibrosis [10,11]. These effects on failing myocardium biology result in an improvement of quality of life and functional capacity [12], reduction of hospitalizations [13], and a biventricular reverse remodeling [14,15] in patients with HFrEF. However, the effects of CCM on the MEE of patients with HFrEF are still unknown; therefore, in this study, we evaluate whether CCM therapy can improve the MEE of patients with HFrEF.

## 2. Materials and Methods

### 2.1. Study Design

We evaluated for inclusion in the study all patients who underwent an Optimizer Smart implant between January 2018 and January 2021 at the Heart Failure Unit of Monaldi Hospital.

The following inclusion criteria were used:(1)left ventricular ejection fraction ≤ 40%,(2)New York Heart Association Class (NYHA) II-IV,(3)Persistence of HF-related symptoms and/or >2 unplanned HF-related visits or hospitalization in the last 12 months despite optimal medical therapy (OMT),(4)QRS duration < 120 ms.

The following exclusion criteria were used:
(1)acute coronary syndrome in the previous three months,(2)cardiac resynchronization therapy device implantation in the previous 12 months,(3)absence of aortic stenosis or left ventricular outflow tract (LVOT) obstruction,(4)non-target dose of OMT for HFrEF,(5)end-stage kidney disease required renal replacement therapy.

During the study period, 27 patients underwent an Optimizer Smart® implant, however, 2 patients died before the six-months follow-up, so the final enrolled population consisted of 25 patients. 

Study data were obtained from all patients 24 h before and six months after CCM therapy. In addition, all patients signed informed consent, the recommendations of the Helsinki Declaration were followed, and the ethics committee of the AORN dei Colli-Monaldi Hospital approved the study (resolution No. 903/2020).

### 2.2. Echocardiography

Standard transthoracic echocardiography and Doppler assessment were performed with Vivid E9 (GE Healthcare, Chicago, IL, USA) as recommended elsewhere [16,17,18]. Three cardiologists with expertise in echocardiography, blinded to this study, acquired and analyzed all echocardiographic images. 

An average of 3 cardiac cycles in patients with sinus rhythm and 5 cardiac cycles in patients with atrial fibrillation was used for the individual measures. According to common practice [19], stroke volume (SV) was calculated as: SV = Left ventricular outflow tract (LVOT) radius^2^ × time velocity integral (TVI) of LVOT.

The global longitudinal strain (GLS) of the left ventricle was measured using the Q-Analysis software package (EchoPAC BT2.02; GE Vingmed, Horten, Norway).

After manually identifying the end-systolic endocardial boundary of the left ventricle by locating three points, a region of interest (ROI) was automatically generated. Next, the ROI was adjusted by the operator in order to include the entire left ventricular walls. Finally, according to international recommendations, we calculated the GLS value as the average of the values obtained from the four chambers, two chambers, and three chambers’ views. The echocardiographic evaluations were performed 24 h before and six months after CCM therapy.

### 2.3. MEE Evaluation

The MEE of a system is the ratio of the work produced to the amount of energy required to produce that work [20]. The MEE of the left ventricle is determined by the ratio of systolic work (SW) to myocardial volume oxygen (MVO2), which expresses the amount of oxygen used by the cardiomyocytes [21].

The following formula were used for calculations:SW = systolic blood pressure (SBP) × stroke volume (SV),
MVO2 = SBP × heart rate (HR),
MEE = SV/HR (where HR is expressed in second, HR/60),
MEEi = MEE/body surface area (BSA).

### 2.4. Statistical Analysis

Prism 9 statistical software (GraphPad Software, San Diego, CA, USA) was used to do all statistical analyses. Clinical and population variables are shown as mean ± standard deviation, and categorical variables are expressed as numbers and percentages. Variations between variables at baseline and follow-up were compared using the Wilcoxon test for variables with nonnormal distribution and the t-test for variables with normal distribution. All *p* values were two-sided; statistical significance was considered for *p* values < 0.05.

## 3. Results

The final study population consisted of 25 patients, whose clinical and echocardiographic characteristics are shown in Table 1.

Most of the patients were male (22; 88%), 13 patients (52%) had an ischemic etiology, and 9 patients (36%) had atrial fibrillation. Additionally, all patients have a previous implantable cardioverter defibrillator, and 7 patients (28%) have a device for cardiac resynchronization therapy.

### 3.1. Effects of CCM Therapy on Left Ventricular Function

The echocardiographic index of left ventricular systolic function improved at the six-months follow-up (Table 2). 

There was a significant left ventricular reverse remodeling with a reduction of end-diastolic (211.8 ± 45.8 vs. 88.3 ± 38.5 mL; *p* = 0.041) and end-systolic volumes (141.8 ± 51.5 vs. 119.6 ± 49.7 mL; *p* = 0.024), with a consequent improvement of left ventricular ejection fraction (30.8 ± 71 vs. 36.1 ± 6.9%; *p* = 0.032). In addition, there was a significant increase in the most specific and reproducible echocardiographic index of left ventricular function, the GLS (−10.3 ± −2.7 vs. −12.9 ± −4.2%; *p* = 0.018; Figure 1). In addition, diastolic function indices also improved, particularly the E/e’ ratio was significantly reduced at six-month follow-up (16.3 ± 7.5 vs. 10.8 ± 4.2; *p* = 0.041).

### 3.2. Effects of CCM Therapy on Natriuretic Peptides, NYHA Class, and Quality of Life

As shown in Figure 2 (panel A) at the six months follow-up, a significant reduction of plasma levels of N-terminal Brian Natriuretic Peptide (NT-proBNP) was observed in the enrolled patients (2975 ± 1988 vs. 1911 ± 1268 pg/mL; *p* = 0.029).

Simultaneously with the reduction of natriuretic peptides plasma levels, an improvement in the symptom reported by the patients occurred; in fact, at follow-up, a statistical reduction in both NYHA class (3.1 ± 0.62 vs. 2.3 ± 0.56; *p* = 0.0001; Figure 2B) and of the Minnesota Living with Heart Failure score occurred (40.08 ± 12.31 vs. 26.9 ± 10.8; *p* = 0.0001—Figure 2C). 

### 3.3. Effects of CCM on MEE

As showed in Figure 3, both MEE (32.2 ± 10.1 vs. 38.6 ± 7.6; mL/s *p* = 0.013) and MEEi (18.4 ± 6.3 vs. 24.3 ± 6.7 mL/s/g; *p* = 0.022) increased after six months of CCM therapy. The improvement of these indexes was due essentially due to the increase of SV without a concomitant increase in HR (Figure 4). From a pathophysiological point of view, this indicates an increase in cardiac contractility in the absence of a corresponding increase in myocardial oxygen consumption, thus leading to an improved mechano-energetic coupling of the heart.

## 4. Discussion

In this study, for the first time, we demonstrate that left ventricular GLS and MEE increased after 6 months of CCM therapy in patients with HFrEF. Longitudinal deformation of the left ventricle is due to the contraction of subendocardial fibers, which are the most susceptible to altered calcium handling [22], increased myocardial stiffness [23], and myocardial fibrosis [24], typical features of the failing heart.

Therefore, longitudinal left ventricular dysfunction and consequentially reduced GLS values develop early in patients with HFrEF [25]. In ex vivo intact hearts, CCM therapy improves calcium handling through several mechanisms, such as rapid normalization of phospholamban phosphorylation [26], upregulation of L-type calcium channels, and increased calcium uptake into the sarcoplasmic reticulum [27]. The latter mechanism results in a rise of extracellular calcium flux during the subsequent cardiac cycle and increased calcium release from the SR itself (the so-called “calcium-induced calcium release”) mechanism [28].

Animal models have demonstrated benefits of CCM therapy. In a canine HFrEF model, CCM therapy reduced left ventricular filling pressure due to the improvement of ventricular compliance and relaxation and improved diastolic Ca^++^ physiology [29]. In a rabbit HFrEF model, CCM therapy reduced cardiac expression of connective tissue growth factor and galectin-3 (a pro-fibrotic marker involved in myocardial structural remodeling) with a reduction of myocardial fibrosis [11]. These effects of CCM therapy observed in animal models may explain the improvement in diastolic function and GLS observed in this study, as well as a reduction of the E/e’ ratio and of the NT-proBNP plasma levels both expression of left ventricular filling pressure.

The improvement in diastolic function justifies the improvement in NYHA class and quality of life observed in patients enrolled in the study. In fact, diastolic function is the main determinant of functional capacity and quality of life in patients with HF [30,31,32], and therefore its improvement is associated with an improvement in these parameters [33].CCM has also been shown to increase stroke volume in a canine HFrEF model [34]; in our study, we documented for the first time that CCM therapy results in an increase in SV at 6 months, even in a population of patients with HFrEF in optimal medical treatment. 

Notably, the improvement in MME observed in our study was caused by an increase in SV without a rise in HR and, consequently, of MVO2. This confirms the findings of a prior study in which CCM increased dP/dt (an index of myocardial contractility) without an increase of MVO2 in nine patients with HFrEF [35]. 

In conclusion, CCM induces an increase of SV and consequently of cardiac output without a concomitant increase in myocardial oxygen demand acting as a smart inotropic therapy.

## 5. Study Limitations

The relatively small number of patients as well as the single-center, observational design of the study with the lack of a control group may influence our results. In addition, although the echocardiographic evaluations were performed in stable patients, the assessments of SV and GLS may be influenced by loading conditions. Seven patients have a CRT-D implanted 12 months before the inclusion in the study; for these patients, late response to this therapy cannot be excluded. 

## 6. Conclusions

At six months of follow-up, CCM therapy increased left ventricular performance, improving left ventricular ejection fraction, E/e’ ratio, GLS, as well as MEE and MEEi in patients with HFrEF on optimal medical therapy.

These echocardiographic improvements are associated with a clear clinical benefit documented by reduction of NT-pro BNP plasma levels NYHA class and MLHFQ score. 

Additional larger studies are needed to provide a greater understanding of the long-term impact of CCM on left ventricular function, as well as the prognostic significance of these observations.

## Figures and Tables

**Figure 1 jcm-11-05866-f001:**
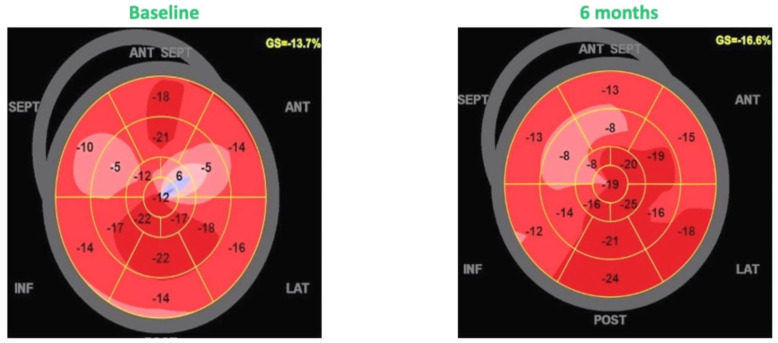
Effects of CCM on global longitudinal strain.

**Figure 2 jcm-11-05866-f002:**
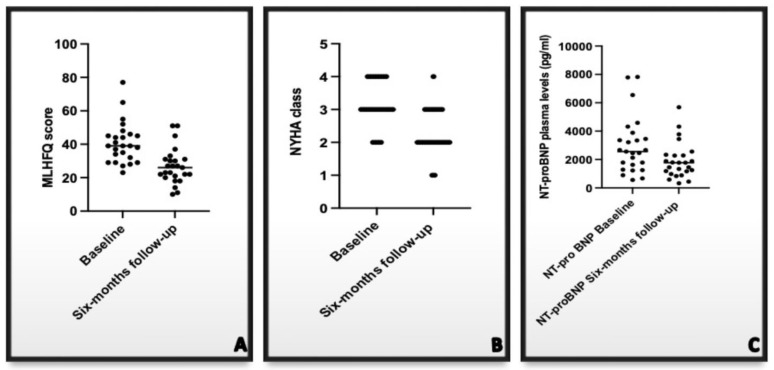
Effects of CCM therapy on NT-proBNP plasma levels (panel (**A**)), NYHA class (panel (**B**)), and MLHFQ score (panel (**C**)). NT-proBNP: N terminal-pro brain natriuretic peptide; NYHA: New York Heart Association; MLHFQ: Minnesota Living with Heart Failure Questionnaire.

**Figure 3 jcm-11-05866-f003:**
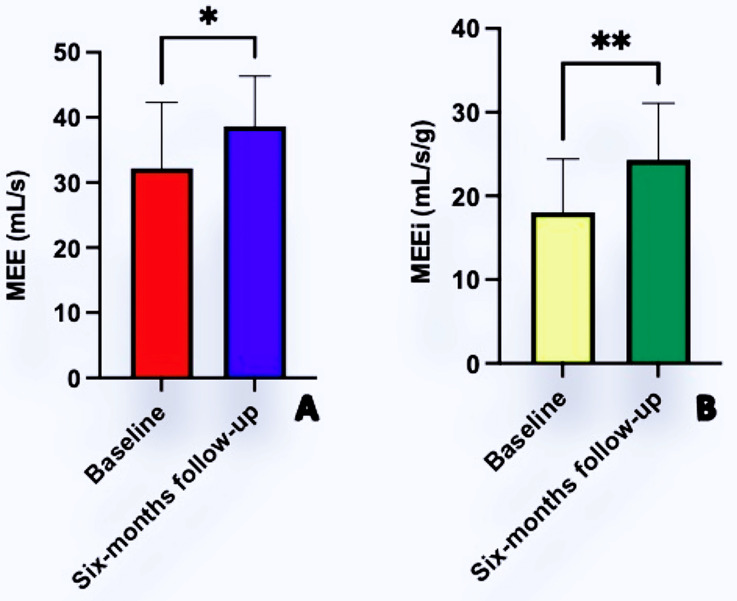
Improvements of Myocardial Mechano-Energetic Efficiency (MEE; Panel (**A**)) and Mechano-Energetic Efficiency index (MEEi; Panel (**B**)) after six months of CCM therapy. * = *p* < 0.05; ** = *p* < 0.001.

**Figure 4 jcm-11-05866-f004:**
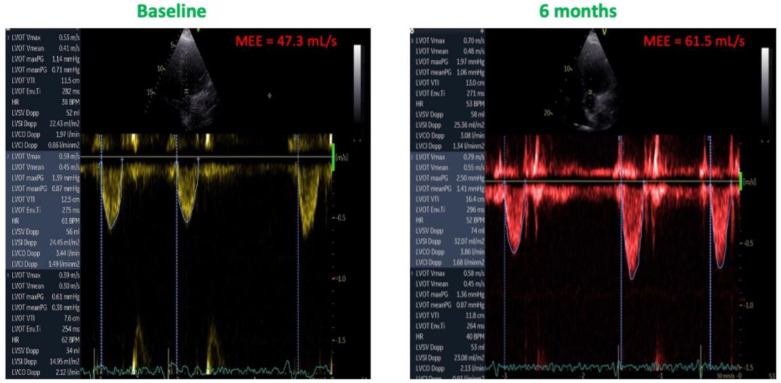
Effects of CCM therapy on MME. Note the increase in stroke volume without an increase in heart rate.

**Table 1 jcm-11-05866-t001:** Clinical and echocardiographic patients’ characteristics at baseline.

Variable	Overall Population (25)
Age (mean ± SD)	62.8 ± 9.7 years
Female sex (n,%)	3 (12%)
Ischemic etiology (n%)	13 (52%)
Hypertension (n, %)	12 (48%)
Diabetes (n,%)	9 (36%)
COPD (n,%)	7 (28%)
NYHA class II (n,%)	4 (16%)
NYHA class III (n,%)	13 (52%)
NYHA class IV (n, %)	8 (32%)
ICD-DR (n,%)	16 (64%)
S-ICD	2 (8%)
CRT-D	7 (28%)
SBP (mean ± SD)	101 ± 11 mmHg
DBP (mean ± SD)	72 ± 6 mmHg
NT-pro BNP (mean ± SD)	2185 ± 1738 pg/mL
e-GFR (CKD-EPI)	62.3 ± 12 ml/min/1.73 m^2^
BUN/Creatinine	18.4 ± 9.7 mg/dL
Atrial fibrillation	9 (36%)
LVEDV (mean ± SD)	208.2 ± 73.2 mL
LVESV (mean ± SD)	125.3 ± 43.5 mL
LVEF (mean ± SD)	32.8 ± 7.1%
LAVi	41.9 ± 4.3 mL/m^2^
E/e’ ratio	16.3 ± 7.5 cm/sec
Loop diuretic (n,%)	16 (64%)
Beta-Blockers (n,%)	25 (100%)
ARNI (n%)	25 (100%)
MRA (n,%)	18 (72%)

COPD: chronic obstructive pulmonary disease; NYHA: New York Heart Association; ICD-DR: dual chamber implantable cardioverter defibrillator; S-ICD: subcutaneous implantable cardioverter defibrillator; CRT-D: cardiac resynchronization therapy with defibrillator back-up SBP: systolic blood pressure; DBP: diastolic blood pressure; NT-pro BNP: N terminal-pro brain natriuretic peptide; e-GFR: estimated glomerular filtration rate; CKD-EPI: chronic kidney disease epidemiology collaboration; BUN: blood urea nitrogen; LVEDV: left ventricular end-diastolic volume; LVESV: left ventricular end-systolic volume; LVEF: left ventricular ejection fraction; LAVi: left atrium volume index; E/e’ ratio: Ratio of mitral peak velocity of early filling to early diastolic mitral annular velocity ARNI: angiotensin receptor-neprilysin inhibitor; MRA: mineral receptor antagonist.

**Table 2 jcm-11-05866-t002:** Echocardiographic index of left ventricular systolic function of the study population.

Variable	Baseline	6 Months Follow-Up	*p*-Value
LVEDV (mL)	211.8 ± 45.8	188.3 ± 38.5	0.041
LVESV (mL)	141.8 ± 51.5	119.6 ± 49.7	0.024
LVEF (%)	32.8 ± 7.1	36.1 ± 6.9	0.032
GLS (%)	−10.3 ± −2.7	−12.9 ± −4.2	0.018

LVEDV: left ventricular end-diastolic volume; LVESV: left ventricular end-systolic volume; LVEF: left ventricular ejection fraction; GLS: global longitudinal strain.

## Data Availability

The data presented in this study are available on request from the corresponding author.

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
