# Peer review of "The Effects of Device-Based Cardiac Contractility Modulation Therapy on Left Ventricle Global Longitudinal Strain and Myocardial Mechano-Energetic Efficiency in Patients with Heart Failure with Reduced Ejection Fraction"

_jcm, 2022, doi:10.3390/jcm11195866_

Round 1

Reviewer 1 Report

Thanks for the chance to review this manuscript. I have two major concerns which I think must be improved;

First, the manuscript does not reach the level of a proper article, it suits more like case series report. I think authors should work on the introduction, improve the discussion and add more details to the results section.

Second, the similarity report is very high. Authors need to work on the writing of the manuscript.

Please write your equation in a proper equation style (as described in the MDPI templates).

Author Response

First, the manuscript does not reach the level of a proper article, it suits more like a case series report. I think the authors should work on the introduction, improve the discussion and add more details to the results section.

Response: We thank the reviewer for this constructive comment. In the new version of the manuscript, we have improved the introduction, results, and discussion sections. 

Second, the similarity report is very high. Authors need to work on the writing of the manuscript.

Response: We thank the reviewer for this constructive comment. We have profoundly revised the manuscript in order to reduce the similarity index

Please write your equation in a proper equation style (as described in the MDPI templates)

Response: We thank the reviewer for this constructive comment. We have rewritten the equations according to Journal style. 

Reviewer 2 Report

The authors examined the effects of CCM therapy on MEE and GLS, and found that CCM could increase left ventricular performance, improving LEVF, GLS, MEE, and MEEi in patients with HFrEF.

Overall the data seemed solid and properly presented, with limitations well acknowledged. I would recommend the following revisions.

1.     Considering the multi-discipline nature of the readership of the journal, a brief introduction of CCM is warranted.

2.     The study would be more complete if routine laboratory results of heart function and HF have been provided and compared.

3.     Similarly, changes of clinical phenome, such as symptoms, signs, and NYHA class, of HF should have been included and compared.

Author Response

The authors examined the effects of CCM therapy on MEE and GLS, and found that CCM could increase left ventricular performance, improving LEVF, GLS, MEE, and MEEi in patients with HFrEF.

Overall the data seemed solid and properly presented, with limitations well acknowledged. I would recommend the following revisions.

  1. Considering the multi-discipline nature of the readership of the journal, a brief introduction of CCM is warranted.

Response: We thank the reviewer for the useful comment.  In the introduction section, we have briefly explained the mechanism of action of CCM.

  1. The study would be more complete if routine laboratory results of heart function and HF have been provided and compared.

Response: We thank the reviewer for the useful comment.  In the revised version of the manuscript, we have added the effects of CCM on NT-proBNP plasma levels.

  1. Similarly, changes of clinical phenome, such as symptoms, signs, and NYHA class, of HF should have been included and compare              Response: We thank the reviewer for the useful comment.  In the revised version of the manuscript, we have added the effects of CCM on NYHA class and MLHFQ score 

Reviewer 3 Report

Dear Sir/Madam,

I had the opportunity to act as a reviewer on the recent submission by Masarone et al. to the Journal of Clinical Medicine.

The authors present original research studying the effect of cardiac contractility modulation therapy on left ventricular GLS and myocardial mechano-energetic efficiency in patients with HFrEF. They found that CCM therapy led to the improvement of left ventricular ejection fraction, GLS, as well as mechano-energetic efficiency.

The manuscript is well structured; however, some issues need to be addressed:

  1. Were LVOT obstruction and aortic stenosis also exclusion criteria? Please comment.
  2. When were the CRT-D devices implanted?
  3. Did the patients receive any sort of AF ablation therapy (PVI) during follow-up?
  4. Line 31: MEE instead of MME?

Best regards,

Author Response

Were LVOT obstruction and aortic stenosis also exclusion criteria? Please comment.

Response: We thank the reviewer for the useful comment. We have specified that aortic stenosis and LVOT obstruction were exclusion criteria.

When were the CRT-D devices implanted?                                                   Response: We thank the reviewer for the useful comment. We have specified that all CRT devices were implanted at least one year before their inclusion in the study.

Did the patients receive any sort of AF ablation therapy (PVI) during follow-up? Response: We thank the reviewer for the useful comment. We have specified that all electrophysiological procedures were performed at least 12 months before the inclusion in the study. However, no patients enrolled in the study underwent AF ablation therapy.

Line 31: MEE instead of MME? 

Response: Correct accordingly  

Round 2

Reviewer 1 Report

Thanks for your effort.

Author Response

We thank the reviewer for their work

Reviewer 2 Report

Thanks my concerns have been addressed

Author Response

We thank the reviewer for their work